# The RUNX Family Defines Trk Phenotype and Aggressiveness of Human Neuroblastoma through Regulation of p53 and MYCN

**DOI:** 10.3390/cells12040544

**Published:** 2023-02-08

**Authors:** Kiyohiro Ando, Akira Nakagawara

**Affiliations:** 1Research Institute for Clinical Oncology, Saitama Cancer Center, Saitama 362-0806, Japan; 2Saga International Carbon Particle Beam Radiation Cancer Therapy Center, Saga HIMAT Foundation, Saga 841-0071, Japan

**Keywords:** neuroblastoma, RUNX, Trk, p53, p63, p73, MYCN

## Abstract

The Runt-related transcription factor (RUNX) family, which is essential for the differentiation of cells of neural crest origin, also plays a potential role in neuroblastoma tumorigenesis. Consecutive studies in various tumor types have demonstrated that the RUNX family can play either pro-tumorigenic or anti-tumorigenic roles in a context-dependent manner, including in response to chemotherapeutic agents. However, in primary neuroblastomas, RUNX3 acts as a tumor-suppressor, whereas RUNX1 bifunctionally regulates cell proliferation according to the characterized genetic and epigenetic backgrounds, including MYCN oncogenesis. In this review, we first highlight the current knowledge regarding the mechanism through which the RUNX family regulates the neurotrophin receptors known as the tropomyosin-related kinase (Trk) family, which are significantly associated with neuroblastoma aggressiveness. We then focus on the possible involvement of the RUNX family in functional alterations of the p53 family members that execute either tumor-suppressive or dominant-negative functions in neuroblastoma tumorigenesis. By examining the tripartite relationship between the RUNX, Trk, and p53 families, in addition to the oncogene MYCN, we endeavor to elucidate the possible contribution of the RUNX family to neuroblastoma tumorigenesis for a better understanding of potential future molecular-based therapies.

## 1. Introduction

Neuroblastomas are the most common extracranial solid tumors of early childhood. The International Neuroblastoma Risk Group (INRG) classification system was developed to categorize neuroblastomas into high, intermediate, and low risk groups in order to allow a consensus for optimal treatment strategies [1]. The representative prognostic variables are stage, age at diagnosis, MYCN oncogene status, chromosome 11q status, DNA ploidy, and the histological category and grade of tumor differentiation. Although intensive multi-modal therapy has improved the overall prognosis, there remains a considerable proportion of patients with unfavorable outcome [2,3]. Conversely, and of particular research interest, some patients with neuroblastoma, especially those under 1 year of age, often experience spontaneous regression of the primary tumor as well as distant metastases in the skin, liver, or bone marrow [4,5].

The Runt-related transcription factor (RUNX) family, which has essential roles in neural development and differentiation through regulation of the tropomyosin-related kinase (Trk) family, also has both oncogenic and tumor suppressive roles in various malignant neoplasms through regulation of the p53 family. Knockout mice studies have indicated that Runx1 down-regulates the number of both TrkA- and TrkC-positive neurons. In contrast, Runx3 up-regulates the number of both TrkA- and TrkC-positive neurons through repression of TrkB signaling (reviewed in [6]). The p53 family members, p63 and p73, possess three major functional domains composed of the NH2-terminal transactivation domain, the central core sequence-specific DNA-binding domain, and the COOH-terminal oligomerization domain. A study on mammalian oocytes suggests that p63 is the most ancient form that then evolved to p73, which is structurally more similar to p63 than p53. Later in evolution, p53 appeared in somatic cells as a tumor suppressor [7]. There are two major NH2-terminally distinct isoforms, a transcriptionally active (TA) and a transcriptionally inactive (∆N) form. Of note, the DNA-binding domain is highly conserved across the family. Therefore, p63 and p73 share the majority of the p53-responsive promoters and have an ability to transactivate a series of p53 target genes implicated in cell cycle arrest and apoptosis (reviewed in [8]). All of the RUNX family members interact with p53 and can influence tumor-suppressive ability [9], whereas RUNX1 contributes to induce protein degradation of p63 and p73 [10,11]. In this review, we integrate functional evidence of the RUNX, Trk, and p53 family members and endeavor to understand a possible tripartite relationship in neuroblastoma tumorigenesis.

## 2. RUNX Family Regulates Cell Survival through Neurotrophin Signals

### 2.1. Neurotrophin Signals in Neuroblastoma Tumorigenesis

It has been well recognized that neuroblastomas originate from aberrant development of sympathoadrenal cells because neuroblastomas preferentially arise in the sympathetic chain or the adrenal medulla as a consequence of the migration and differentiation of neural crest cells (NCCs) [12,13]. Neuroblastoma tumorigenesis is closely related to the expression of the neurotrophins and their specific receptors, the Trk family; TrkA, TrkB and TrkC (also known as NTRK1, NTRK2, and NTRK3, respectively). The preferred ligands for TrkA, TrkB, and TrkC are nerve growth factor (NGF), brain-derived neurotrophic factor (BDNF) and neurotrophin-4 (NT-4), and neurotrophim3 (NT-3), respectively. During development, the Trk family regulates the MAPK, PI3K, and PKC pathways to mediate survival and differentiation of both central and peripheral neurons. Importantly, expression of TrkA and TrkC is positively correlated with favorable biological features of neuroblastoma [14,15], whereas increased expression of TrkB and BDNF is associated with an unfavorable prognosis in patients concomitant with MYCN-amplification [16,17]. In contrast, an autocrine loop of the NGF/TrkA is associated with protumorigenic activity in both breast and prostate carcinomas [18,19]. Similarly, increased co-expression of NT-3 and TrkC are observed in a subset of stage IV neuroblastomas and high-grade gliomas [20,21]. These findings are possibly linked with the idea that TrkA and TrkC behave as dependence receptors, which trigger apoptosis when unbound by their respective ligands [22]. Somatic mutations of the Trk family have been identified in various tumor types. However, the role of each Trk overexpression and mutation in promoting tumorigenesis and cancer progression has been reported as variable with either accelerating or inhibiting effects. Most notably, there is increasing evidence that oncogenic translocation of Trk family members is associated with morbidity and mortality, as well as clinical efficacy of targeted therapeutics (e.g., NTRK inhibitors) in multiple tumors from both adult and pediatric patients (Table 1) [23,24,25]. These chromosomal rearrangements involve fusing of the kinase domain of a Trk family member to an unrelated gene instead of its regulatory domain, yielding constitutive kinase activation. In neuroblastoma, a ligand-independent oncogenic variant of TrkA, named TRKAIII, that lacks one of two functional extracellular immunoglobulin regions (IG-C1) and N-glycosylation domains has been identified [26]; however, reports of pathogenic mutations of the Trk family are otherwise scarce. Intriguingly, primary neuroblastoma cells with high TrkA expression display differentiation with neurite outgrowth in the presence of NGF but undergo apoptosis in the absence of NGF, giving insight into the mechanisms underlying spontaneous regression of neuroblastomas [14]. This suggests that ligand deprivation could trigger spontaneous regression of neuroblastomas, phenocopying the developmental process in sympathoadrenal cells. Furthermore, TrkA-induced apoptosis of neuroblastoma cells is mediated, at least in part, through p53 function [27].

In summary, neurotrophin signals orchestrated by the Trk family are closely related to neuroblastoma tumorigenesis through, at least in part, the alteration of their expression. However, the lack of Trk family mutations identified in neuroblastomas, despite intensive searches for genomic and transcriptomic alterations, suggests that Trk family expression may be altered by other pathways, for example dysregulation of upstream regulators such as the RUNX family.

### 2.2. RUNX Family Regulates Trk Family in Neuroblastoma

NCCs are multipotent progenitor cells that have the ability to differentiate into several cells other than those in the sympathoadrenal linage, including the nervous sensory, mesenchymal, melanocyte, osteoclast, and chondrocyte lineages. A direct relationship between the Trk and RUNX families has emerged from studies on the development of the dorsal root ganglion (DRG), in which somatosensory inputs are transmitted from the periphery to the spinal cord [6].

Runx3-deficient (Runx3(−/−)) mice display severe motor incoordination that is causally related to defects in TrkC-positive proprioceptive afferents [28,29]. Subsequent studies demonstrated that RUNX3 determines TrkC-positive neuron identities in the presence of NT3 (the ligand for TrkC) through the transcriptional repression of TrkB in order to segregate from TrkBTrkC double-positive neurons [30,31]. Meanwhile, possible involvement of the Runx family in lineage commitment of the NT4-dependent TrkB-positive neurons remains elusive [30], although mice deficient in NT-4 (the ligand for TrkB) lose TrkB-expressing neurons, with a concomitant increase in TrkC-positive neurons [32] (Figure 1A). Additionally, the number of TrkA-positive neurons is decreased in Runx3(−/−) mice [33], indicating that Runx3 positively regulates not only TrkC but also TrkA, possibly in an indirect fashion (Figure 1B).

In contrast, Runx1 deletion in mice (Runx1(−/−)) is embryonic lethal due to defective hematopoiesis. Notably, although conditional knockout mice lacking Runx1 in premigratory NCCs show no overt abnormalities in the number of DRG neurons, they display a drastic increase in the percentage of TrkA-positive neurons with a concomitant decrease in Ret-positive neurons. This indicates that Runx1 is required for the developmental transition to a Ret-positive neuron identity by repressing TrkA in the peripheral nervous system [34]. Furthermore, transgenic Runx1(−/−) mice, in which hematopoietic cells are selectively rescued, show an increase in both TrkA- and TrkC-positive neurons [35], suggesting that Runx1 has the opposite impact on regulation of the number of TrkA- and TrkC-positive neurons to that of Runx3 (Figure 1B).

In summary, considering an increase in TrkA and TrkC expression is associated with favorable biological features in neuroblastoma [6,7], RUNX3 may simply be categorized as a functional tumor suppressor in neuroblastoma because of its potential to positively regulate TrkA and TrkC. In contrast, RUNX1-mediated downregulation of TrkA and TrkC implies that RUNX1 has oncogenic potential in neuroblastomas. Consistently, loss of RUNX3 was associated with an increase in TrkB signaling, which marks an unfavorable prognosis in patients with neuroblastoma [30]. Therefore, the differential impact of neurotrophin signals in the biology of neuroblastomas, as well as patient prognosis, might be closely related to the pattern of expression of the RUNX family members (Figure 1C).

## 3. RUNX Family Regulates Cell Survival through p53-Family-Related Pathways

### 3.1. Aberration of TP53 Family-Related Pathways in Neuroblastoma

Although more than 50% of human cancers possess mutations in TP53 that alter their transcriptional activity, the frequency of TP53 mutations is relatively rare in neuroblastoma. The mutation rate is less than 2% in primary neuroblastomas, increasing to 15% following chemotherapy and/or at relapse. Importantly, abnormalities in the p53/murine double minute 2 (MDM2)/p14ARF (hereafter, ARF) pathway are reported in almost half of cases. In particular, genomic amplification of MDM2 and promoter-methylation/chromosomal-deletion-mediated inactivation of ARF are present at both diagnosis and relapse in the majority of cases, implying a critical role for the p53-related pathway in neuroblastoma development [36] (Figure 2). Consistently, MYCN binds directly to a noncanonical E-Box DNA binding motif located upstream of the transcriptional start site within the p53 promoter and upregulates p53 transcripts in neuroblastoma (Figure 2). Therefore, MYCN-induced p53 is recognized as a safeguarding mechanism preventing aberrant proliferation in the normal embryonic development of the nervous system [37]. It is quite similar to the observation that reintroduction of oncogenic RAS in primary human or rodent cells results in cellular senescence through accumulation of p53. [38] In addition, the transformation from rodent fibroblasts does not occur with transfection of Ras oncogenes alone unless there is a cooperating oncogene such as *Myc*, another member of the MYC family (of MYCN) [39].

Initial studies have provided evidence that p53 activity is lacking in many neuroblastomas, for instance, by cytoplasmic sequestration of wild-type p53 in undifferentiated neuroblastomas that leads to impaired G1 arrest in response to DNA damage [40]. The G1 arrest was shown to be mediated by cyclin-dependent kinase inhibitor 1A (CDKN1A, also known as p21^*CIP*^^/*WAF1*^, hereafter p21), a representative transcriptional target of p53. Briefly, p21 has the ability to inhibit G1-cyclin-dependent kinases, blocking cell cycle progression from G1 into S phase. Another mechanism of p53 inactivation in neuroblastomas is overexpression of MDM2. In neuroblastoma, MDM2 amplification is relatively common and causally related to inhibition of p53 function. Mechanistically, the N-terminus of MDM2 binds to p53 and represses p53-mediated transcription, whereas the C-terminus acts as an E3 ubiquitin ligase that mediates degradation of p53 (Figure 2). Therefore, MDM2 is widely established as a representative negative regulator of p53. Although a number of preclinical investigations have assessed antagonists targeting the p53–MDM2 interaction in neuroblastoma, the clinical efficacy of these agents still has not been fully defined [41].

Another member of the p53 family, p73, functionally overlaps with p53 in its ability to induce cell cycle arrest and/or apoptosis, therefore also acting as a tumor suppressor. However, in contrast to p53-deficient mice, those lacking p73 do not show increased susceptibility to spontaneous tumorigenesis but exhibit neurological and immunological defects [42]. Activation of p73 isoforms with a functional transactivated domain (hereafter, TAp73) in human cancer in response to chemotherapeutic regents induce apoptosis even in cells lacking p53. A p53 mutant protein can interact with TAp73, resulting in an impaired ability for apoptotic induction [43]. The 1p36.2-3 chromosome region, in which the p73 gene is located, is heterozygously lost in a variety of human cancers including neuroblastoma; however, few p73 mutations have been reported despite an extensive search [44]. It is presently unclear whether p73 could be an essential tumor suppressor gene in neuroblastoma. In contrast, ΔNp73, which is one of the isoforms of p73 lacking the N-terminal transcriptional activated domain, is recognized as an oncogene, conveying a dominant-negative effect on p53 and TAp73 (Figure 2). Notably, ΔNp73 may broaden our understanding of the possible link between p53 inactivation and neuroblastoma tumorigenesis. The p73−/− mouse, which lacks all isoforms of p73, shows increased apoptosis in developing sympathetic neurons, indicating that ΔNp73 inhibits neuronal programmed cell death by blocking the pro-apoptotic function of p53 [45]. Consistent with its anti-apoptotic function, the increased levels of ΔNp73 expression are significantly associated with poor survival in patients with neuroblastomas, whereupon it predicts worse outcome independently of other risk factors such as patient age and MYCN amplification [46].

Previously, we reported that TAp63, which is one of the isoforms of p63 containing the N-terminal transcriptional activation domain, has the ability to repress transcription of the MYCN and NCYM genes; the latter being a cis-antisense gene of MYCN responsible for stabilizing MYCN protein and accelerating distant metastases in neuroblastoma (Figure 2). Further, all-trans-retinoic acid (ATRA) treatment of the neuroblastoma cell line CHP134 predominantly induced expression of TAp63 but decreased expression of p53 and TAp73; meanwhile, the expression of MYCN is downregulated by TAp63 directly and indirectly through NCYM [47]. Therefore, it has been proposed that ATRA treatment induces TAp63 expression leading transcriptional repression of MYCN expression. Subsequently, the down-regulation of MYCN results in inhibition of the expression of its transcriptional targets including the tumor suppressor p53, indicating that differential functions among the p53 family may be causally related to the conflicting cellular responses to ATRA treatment such as promoting neuronal differentiation and/or apoptosis in neuroblastomas. Since ATRA treatment has potential clinical application for the treatment of neuroblastoma, it is consistent that the increased expression levels of TAp63 mRNA are associated with a favorable prognosis in patients with neuroblastomas [47].

### 3.2. Role of RUNX1 and the p53 Family in Neuroblastomas

Aberrations of the RUNX1 gene ascribed by chromosomal translocation and mutations are frequently detected in hematological malignancies, implying that RUNX1 acts as a classic tumor suppressor. Although decreased levels of RUNX1 expression are associated with metastasis in a wide range of cancers, including those arising from the lung, breast, prostate, colorectal, uterus, and ovary [48], increased RUNX1 expression levels might also be involved in oncogenesis in certain types of cancer. For instance, RUNX1 gene expression is upregulated in endometrioid carcinomas and correlates with the sequence of tumorigenesis, from normal atrophic endometrium, to hyperplasia, and then on to carcinoma [49]. Further, RUNX1 loss impairs tumor initiation and maintenance and the growth of oral, skin, and ovarian epithelial human cancer cells through inhibition of the JAK/STAT pathway [50].

There are also some conflicting findings regarding the role of RUNX1 in neuroblastoma tumorigenesis. Increased expression levels of RUNX1 are correlated with favorable overall survival and are detected in ganglioneuromas and well-differentiated neuroblastomas, whereas lower levels are present in poorly differentiated, undifferentiated, and non-MYCN amplified neuroblastomas, indicating RUNX1 as a favorable prognostic indicator [51]. Moreover, athymic nude mice bearing xenografts formed by subcutaneous injection of neuroblastoma SH-SY5Y cells stably expressing RUNX1 or short hairpin RNAs of RUNX1 display inhibited or accelerated tumor development, respectively. These findings are consistent with the reports that RUNX1 overexpression induces cell death as well as cell cycle arrest or differentiation in neuroblastoma cell lines. Further, RUNX1 was proposed to induce apoptosis by increasing transcription of CSF2RB and NFKBIA and decreasing BIRC5 transcription. [51] However, an initial study revealed that competitive inhibition of endogenous RUNX1 function by transfection of AML1a, a naturally occurring gene product of a RUNX1 splice variant, induced cell death in most neuroblastoma cell lines, suggesting that physiological levels of RUNX1 may be essential to maintain continuous neuroblastoma cell growth [52]. This finding is reminiscent of the pivotal role played by the RUNX genes in the dorsal root ganglion (DRG) cell lineage [6]. In addition, excessive expression of RUNX1 under certain experimental conditions may functionally mimic tumor-suppressive RUNX3 (as described later) depending on the similarity of the consensus sequence for DNA binding, suggesting the presence of functional redundancy [52].

There is some mechanistic insight into the functional relationship with p53-family-related pathways that suggests how RUNX1 has, at least in part, the potential to be associated with favorable or unfavorable prognosis in neuroblastomas. The p300 histone acetyltransferase and CREB-binding protein (CBP) interact with RUNX1 and act as transcriptional coactivators during the differentiation of myeloid cells [53]. On the other hand, p300-mediated acetylation enhances the transcriptional activity and the transforming potential of RUNX1 [54]. Based on the binding ability between RUNX1 and p300, we previously found that RUNX1 facilitated p53-mediated transcriptional activity via a complex formation between p53 and p300 in p53-proficient osteosarcoma U2OS cells and HCT116 colon cancer cells treated with adriamycin (ADR) [55]. The p300/CBP3 family of acetyltransferases mediates acetylation of p53 at Lys-373/Lys-382, stimulating its sequence-specific DNA-binding activity [56]. Therefore, it is plausible that increased expression of RUNX1 enables neuroblastoma cells, which possess functional wild-type p53, to undergo apoptosis in response to a chemotherapeutic agent (Figure 3A). However, there is conflicting evidence that, in response to DNA damage, RUNX1 is associated with stabilization of ΔNp73 (as described later), which has the ability to eliminate p53-mediated neuronal apoptosis [45] (Figure 3A). Taken together with these findings, it is reasonable that the chemotherapeutic-agent-induced p53 activation is accomplished, at least in part, through cooperation with RUNX1, whereas RUNX1-mediated stabilization of ΔNp73 is immediately initiated as a fail-safe mechanism to prevent detrimental effects caused by the excessive p53 activation.

Homozygous deletion of the ARF-INK4a locus in chromosome 9 occurs frequently in a variety of human cancers, suggesting that representative tumor suppressors are located in this lesion [57,58]. Mechanistically, ARF binds to MDM2 and promotes MDM2 degradation, resulting in p53 stabilization and accumulation [59]. In its tumor-suppressive role, RUNX1 is known to negatively regulate the ARF transcript by the aberrant chromosome translocation-mediated fusion protein of RUNX1. One of the most frequent translocations, t(8; 21), representing 12–15% of patients with AML, results in fusion of the DNA-binding domain of RUNX1 (also known as AML1) to the eight-twenty-one (ETO, also known as MTG8) co-repressor. Notably, AML1–ETO, as a dominant negative oncoprotein, repressed the ARF promoter and reduced endogenous levels of ARF expression, indicating that RUNX1 functions upstream of the tumor suppressive ARF–p53 pathway [60]. Considering that abnormalities in the p53/MDM2/ARF pathway are identified in half of neuroblastoma cases, RUNX1-mediated ARF upregulation resulting in p53 stabilization may function in at least some of the remaining neuroblastoma cases (Figure 3A).

Notably, studies on the roles of p73 and p63 have highlighted a possible function of RUNX1 in regulating neuroblastoma cell proliferation. RUNX1 regulates the protein stability of p73 through binding to the promotor of Itch, one of the known E3 ligases for p73 [11]. Specifically, RUNX1 interacts with Yes-associated protein (Yap1) and collaborates to induce the transcriptional activation of Itch under normal conditions, sustaining the low levels of p73 protein. However, in response to DNA damage, this interaction is prevented by the non-receptor tyrosine kinase c-Abl through the phosphorylation of Yap1 at Tyr-357, leading to stabilization of the p73 protein [61,62]. Importantly, Itch can also facilitate proteasomal degradation of ΔNp73 [11] (Figure 3A). As mentioned earlier, considering that ΔNp73 inhibits neuronal programmed cell death by blocking the pro-apoptotic function of p53 [45], DNA damage by a chemotherapeutic agent or decreased expression levels of RUNX1 may confer ΔNp73-mediated anti-apoptotic ability on neuroblastomas that express wild-type p53. In addition, ΔNp63, a dominant negative isoform of p63, directly regulates RUNX1 expression for proper differentiation of keratinocytes in the human interfollicular epidermis [63] (Figure 3A). Currently, the relationship between ΔNp63 and neuroblastoma tumorigenesis is still unclear; however, ΔNp63 promotes tumor growth in neuroblastoma xenografts with concomitant tumor angiogenesis [64]. Interestingly, Itch can also facilitate proteasomal degradation of ΔNp63 [10], indicating that RUNX1 deficiency in neuroblastomas may contribute to cell proliferation also through stabilization of ΔNp63. In contrast, as described earlier, MYCN is downregulated through TAp63-mediated downregulation of NCYM [47]. Therefore, functional interruption of RUNX1 that is causally related to stabilization of TAp63 may potentially contribute to preventing MYCN-driven cell proliferation of neuroblastoma (Figure 3A).

Collectively, several lines of evidence indicate that RUNX1 has the potential to affect p53-family-related pathways, resulting in both cell proliferation and cell death in neuroblastomas. Importantly, interruption of RUNX1 function triggered by its expression and DNA damage has the potential to stabilize all of the oncogenic or tumor-suppressive p73 and p63 protein isoforms through downregulation of Itch. Therefore, RUNX1 function is largely dictated not only by tissue specificity (ΔNp73 is predominantly expressed in the human corpus callosum [65]) as well as developmental process (fetal tissues express 10-fold more p73 [both TA and ΔN] than the corresponding adult tissues [66]) but also the presence of functional wild-type p53. Investigation of cell dependency on the p53 family in individual neuroblastomas may provide us with a better understanding of why RUNX1 appears to act as a two-faceted regulator for cell survival.

### 3.3. Role of RUNX2 and the p53 Family in Neuroblastomas

RUNX2 is essential for osteoblast development, but its role in neuronal development is still poorly understood. However, there is a potential link between RUNX2 and p53 or MYC that may provide a mechanism by which RUNX2 could regulate cell survival in neuroblastoma. Primary Runx2-null osteoblasts have a growth advantage and fail to undergo senescence owing to a loss of p21 and ARF expression. Therefore, RUNX2 is suggested as a tumor suppressor in osteoblast progenitors [67]. Curiously, it has been shown that RUNX2 can bind to histone deacetylase 6 (HDAC6) and inhibit the transcription of p21 in mature osteoblast lineage cell lines [68], suggesting that RUNX2 may have a context-dependent function. HDACs are multi-protein complexes that are recruited onto specific DNA elements by various transcriptional factors resulting in gene repression. We previously found that the RUNX2/p53 complex is recruited onto p53 target promoters in osteosarcoma U2OS cells treated with ADR, leading to inhibition of the DNA-damage-induced transcriptional activity and pro-apoptotic activity of p53 [69] (Figure 3B). In addition, depletion of RUNX2 expression improves gemcitabine cytotoxicity in the pancreatic cancer cell lines AsPC-1 (p53-negative) and MiaPaCa-2 (p53-mutated pancreatic cancer), potentially through stimulation of TAp63- and TAp73-dependent cell death, respectively [70,71]. Conversely, p53 could abrogate transcription of RUNX2 through inducing microRNA-34c (miR-34c) [72], in accordance with a finding that remarkable expression of RUNX2 is observed in p53−/− osteoblast progenitor cells [73].

The pro-oncogenic potential of RUNX2 is supported by several findings: chromosome amplification of the genomic region containing RUNX2 is found in osteosarcoma [74] and increased expression of RUNX2 is associated with poor chemo-sensitivity in osteosarcomas [75], progression of prostate cancer in patients [76], and metastasis in breast cancer MDA-MB-231 cells [77]. Notably, a strong link between *Runx2* and *Myc* has been elucidated in mice lymphomagenesis. Briefly, *Runx2* neutralizes the pro-apoptotic effect of *Myc* overexpression and confers a potent survival advantage to thymic lymphomas in *Runx2*/*Myc*-ER^TM^ mice (CD2-*Runx2* mice crossed with CD2-*Myc*-ER^TM^ mice). Importantly, *Runx2*/*Myc* tumors show the low apoptotic phenotype even when a functional p53 is retained in vivo [78].

Collectively, RUNX2 could possibly contribute to neuroblastoma tumorigenesis by collaborating with MYCN as well as by its inhibitory effect on the p53 family (Figure 3B).

### 3.4. Role of RUNX3 and the p53 Family in Neuroblastomas

A chromosomal deletion of the major tumor-suppressor center 1p36-35, in which RUNX3 is epigenetically inactivated, is frequently present in diverse cancer types, including 20% to 40% of cases of patients with neuroblastoma [79]. Accordingly, decreased expression of RUNX3 mRNA is significantly correlated with a partial deletion at chromosome 1p36 as well as poor survival rate, arguing for a tumor suppressor role for RUNX3 in neuroblastoma [80]. The epigenetic inactivation of RUNX3 will be described later in detail.

One mechanistic function of RUNX3 in neuroblastoma is its facilitation of the proteasomal degradation of MYCN [80] (Figure 4A), whereas several other underlying mechanisms have been suggested for its tumor suppressor activity in other type of cancers. In the gastric epithelium, RUNX3 is a downstream target of TGF-β-induced cell cycle arrest and apoptosis through cooperation with SMADs and transcriptional upregulation of the BCL-2-interacting mediator of cell death (BIM), respectively [81,82]. However, we reported that RUNX3 is involved in ADR-mediated phosphorylation of p53 at Ser-15 and enhanced p53-mediated transcriptional and pro-apoptotic activity in U2OS cells [83]. Mechanistically, RUNX3 recruits phosphorylated forms of ATM onto p53 and induces ATM-dependent phosphorylation of p53 at Ser-15 after DNA damage, suggesting that RUNX3 acts as a co-activator for tumor-suppressive p53 (Figure 4A). Furthermore, the clinical outcome of patients with lung adenocarcinoma showed that coexistence of the decreased RUNX3 expression and p53 mutation defined the worst prognosis, as compared with either single alteration. Of relevance to this, disruption of Runx3 by intranasally infecting Runx3^f/f^ mice with adeno-Cre induces adenomas (AD) in the lung and accelerates Kras^G12D^-driven adenocarcinoma (ADC) development as effectively as loss of p53, indicating that loss of RUNX3 is sufficient to override the effect of KRAS/oncogene-induced senescence in AD formation [84]. Conversely, p53 could decrease transcription of RUNX3 through inducing microRNA-34a (miR-34a) [85], similar to the transcriptional regulation of RUNX2 abrogated by the p53-mediated increase in miR-34c [72] (Figure 4A). Interestingly, like RUNX1, exogenous expression of RUNX3 has the ability to increase the expression of ARF, p53, and p21, indicating that RUNX3 is considered as an upstream regulator of the p53 tumor suppressor network [84]. Moreover, regulatory regions of the p21 gene also possess multiple RUNX binding sites, suggesting the possible mechanism of direct regulation of p21 by RUNX3 [81]. However, RUNX3-mediated regulation of the p53 family in neuroblastoma initiation/development remains to be elucidated. Taken together, RUNX3 is advocated as a potential tumor suppressor, both in p53-dependent and p53-independent manners. Therefore, decreased expression of RUNX3 due to deletion of the 1p36 region confers an advantage to a driver oncogene, at least in part, for sustainable proliferation.

Lastly, several lines of evidence for the oncogenic potential of RUNX3 are indispensable for understanding its two-side nature. For example, a possible relationship between mutant p53 and RUNX3 was found in a study of a mouse model of pancreatic ductal adenocarcinoma (PDA) that was genetically engineered to concomitantly express Trp53^R172H^ and Kras^G12D^ in the pancreas. This PDA mouse model displayed markedly elevated RUNX3 transcript levels, which were shown to increase the migratory and metastatic potential of the PDA cells [86]. Considering that wild-type p53 has the ability to abrogate RUNX3 expression through inducing miR-34a [85], it is plausible that the gain-of-function mutation of Trp53^R172H^ (codon 175 in humans), which recapitulates a germline mutation from Li–Fraumeni Syndrome [87], might not only enhance RUNX3 transcription but also convert the function of RUNX3 from tumor suppressive to dominant oncogenic. This reminds us of the basic observation that exogenous expression of RUNX3 results in increased phosphorylation of p53 on Ser15 [83], which is responsible for not only inducing its transactivation but also preventing MDM2-mediated degradation. This contradictory role is supported by a cohort study that demonstrated that high RUNX3 levels in patients with primary pancreatic cancer were correlated with worse survival after resection [86]. Since then, several lines of evidence implicate inhibition of p63 or p73 through direct binding as a potential oncogenic mechanism for the gain-of function mutation of p53 [88,89]. RUNX3 may be causally related to stabilization as well as the synergistic activation of mutant p53, resulting in inhibition of the tumor-suppressive function of the other p53-family members. The answer to the question of whether the point mutant TP53^R175H^ could form stable complexes with RUNX3 is eagerly anticipated.

In summary, certain levels of RUNX3 expression may be required for proper activity of the p53 family in response to chemotherapeutic agents in neuroblastoma. However, due to the rarity of p53 mutation in primary neuroblastomas, it is unclear whether RUNX3 could behave as an oncogene.

## 4. RUNX3 Involvement in Regulatory Circuit in Neuroblastoma Epigenomics

Aberrant DNA methylation has been extensively studied in various tumors and recognized as one of the hallmarks of epigenetic dysregulation in tumorigenesis. Frequent and concurrent hypermethylation of CpG islands in multiple genes, designated as the CpG island methylator phenotype (CIMP), is associated with prognosis in various tumors. Importantly, loss of CIMP-associated genes is sufficient to confer a cellular context that avoids senescence and undergoes transformation by BrafV600E on normal colon-derived organoids of mice [90,91]. In neuroblastomas, the protocadherin-β (PCDHB) family is specifically methylated in patients with poor prognosis. Even among neuroblastomas without MYCN amplification, CIMP was a significant prognostic indicator for patients with poor outcome [92]. Various genes have been identified as markers to define CIMP in different tumor types, and RUNX3 is one of the five best marker genes for CIMP determination in colorectal cancers [93].

Epigenetic inactivation of RUNX3 is tightly linked to the early stages of bladder, lung, and gastrointestinal carcinogenesis, suggesting that RUNX3 is a potential tumor-suppressor. The epigenetic inactivation of RUNX3 expression, concomitant with the chromosome loss of 1p36 in neuroblastomas, is mediated at least in part by the enhancer of zeste 2 (EZH2) [94] (Figure 4B). EZH2 is a methyltransferase functioning in gene silencing through tri-methylation of histone-3 on lysine-27 (H3K27). EZH2 is the catalytic subunit of the Polycomb repressor complex 2 (PRC2). Although essential for stem cell identity and pluripotency, PRC2 also displays oncogenic potential when localized to tumor-suppressor gene loci. In addition, CASZ1, another candidate tumor-suppressor in neuroblastomas harboring 1p36, is also epigenetically inactivated by EZH2. CASZ1 is a zinc-finger transcription factor and regulates a series of gene sets involved in cell proliferation. Of note, p75^NTR^ (also known as NGFR), which is a representative marker of favorable prognosis in patients with neuroblastoma, is also silenced by EZH2. Consistently, an increase in EZH2 expression is significantly associated with poor outcome in patients with neuroblastoma [94]. Furthermore, the promoters of TrkA are repressed not only by DNA methylation but also directly by EZH2-mediated histone modification in another promoter region [95] Since both MYCN and MYC directly induce EZH2 expression through binding at its promoter region [96,97], EZH2-mediated inactivation of the tumor suppressors in 1p36 may conceivably be a major route toward development of unfavorable neuroblastomas with MYCN amplification (Figure 4B).

## 5. RUNX3 Acts a Negative Regulator of the MYCN Protein in Neuroblastomas

Based on a study using neuroblastoma-derived cell lines, even the transactivation deficient mutant form of RUNX3 promotes ubiquitination and protein degradation of MYCN, suggesting that RUNX3 might recruit an E3 ubiquitin ligase of MYCN in a transactivation-independent manner (Figure 1C and Figure 4). However, the E3 ligase responsible for this process is still unknown. Consistently, several patients categorized in the high-MYCN mRNA expression group who survived neuroblastoma concomitantly possessed high mRNA expressions of RUNX3, suggesting that RUNX3 has an ability to overcome the aggressive behavior of MYCN [80]. Interestingly, EZH2 directly interacts with MYCN and induces the protein stabilization of MYCN by competing against FBW7α, a known E3 ligase of the MYC family, in a methyltransferase-independent manner [98] (Figure 4B). Considering EZH2 as a potential epigenetic repressor of RUNX3, there could be a regulatory mechanism for the MYCN protein by mutually exclusive binding to oncogenic EZH2 or tumor-suppressive RUNX3 in neuroblastomas. In contrast to MYCN, MYC has bidirectional transcriptional activation with RUNX3 in hematological malignancies [99,100]. Further study on the detailed mechanism through which RUNX3 transcripts are regulated by MYC and MYCN is warranted.

## 6. Conclusions

Accelerated proliferation induced by an oncogene requires, at least in part, a safeguarding mechanism to prevent deleterious consequences such as mitotic catastrophe [101]. Therefore, direct induction of p53 by MYCN could be a plausible mechanism for tumor cell proliferation, similar to that in the normal embryonic development of the nervous system. In this regard, certain levels of p53 or TAp73 may have crucial roles in neuroblastoma proliferation. The dual, pro-tumorigenic and tumor-suppressive functions of RUNX3 are suggested to be closely related to the regulation of p53 and MYC [102]. In this proposed model, p53 deficiency may convert RUNX3 to an oncogene, resulting in aberrant upregulation of MYC. However, in neuroblastoma, RUNX3 may be essential to counteract MYCN but possibly also to activate p53, therefore potentially acting as a conventional tumor suppressor. Moreover, the RUNX3 deficiency attributed to deletion of the 1p36 region leads to worse prognosis in patients with neuroblastoma via TrkB. Since the deletion of 1p36 may result in deficiency of all p73 family members, the defect in ΔNp73 function may be causally related to activated p53. As mentioned above, p53 activation is rather favorable for MYCN-driven cell proliferation. Collectively, deficiency of both RUNX3 and p73 caused by deletion of the 1p36 region is likely to provide a favorable environment for MYCN-driven neuroblastomas. In fact, the 1p36 deletion is strongly correlated with MYCN amplification in patients with neuroblastoma [103]. In this situation, TAp63 could be particularly important for reducing tumor aggressiveness by decreasing MYCN transcripts [47]. Most notably, not only p53 activation but also TAp63 degradation is, at least in part, responsible for RUNX1. Therefore, in contrast to RUNX3, RUNX1, together with the RUNX1-mediated decrease in TrkA and TrkC, may be essential for neuroblastoma proliferation, especially in neuroblastomas with the 1p36 deletion. Similarly, in MYCN-nonamplified neuroblastomas, targeting the RUNX1-mediated p53 activation may be effective for abolishing their fail-safe mechanism because of the presence of other oncogenes. Conversely, the p53 non-functional neuroblastomas may rely on TAp63 and TAp73 for their survival, suggesting that the reactivation of RUNX1 is a possible therapeutic option (Figure 4C). Further investigation of targeting RUNX1 may be advantageous as a potential therapeutic strategy for neuroblastoma.

## Figures and Tables

**Figure 1 cells-12-00544-f001:**
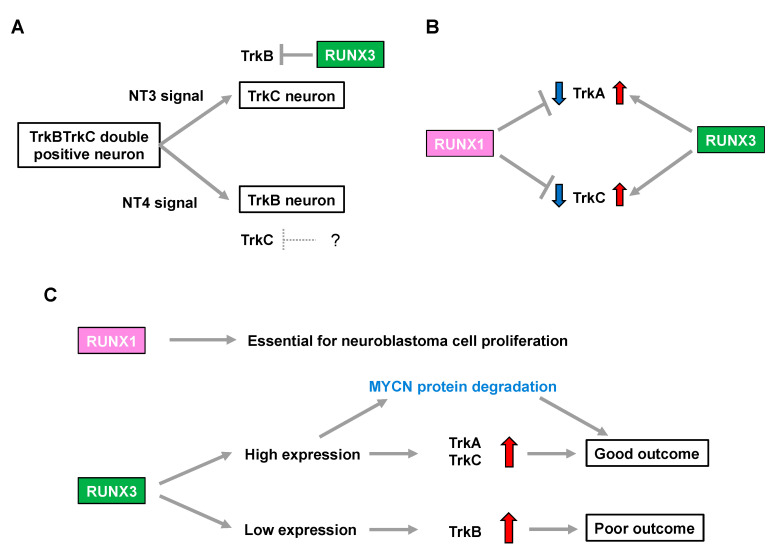
RUNX family is associated with neuroblastoma aggressiveness through regulation of Trk family expression. (**A**) Neurotrophin signals define the cell fate in development of the dorsal root ganglion. The question mark indicates a hypothetical repressor of TrkC. (**B**) RUNX1 and RUNX3 have opposite impacts on regulation of TrkA and TrkC expression. (**C**) RUNX 1 and RUNX3 influence neuroblastoma biology and patient prognosis.

**Figure 2 cells-12-00544-f002:**
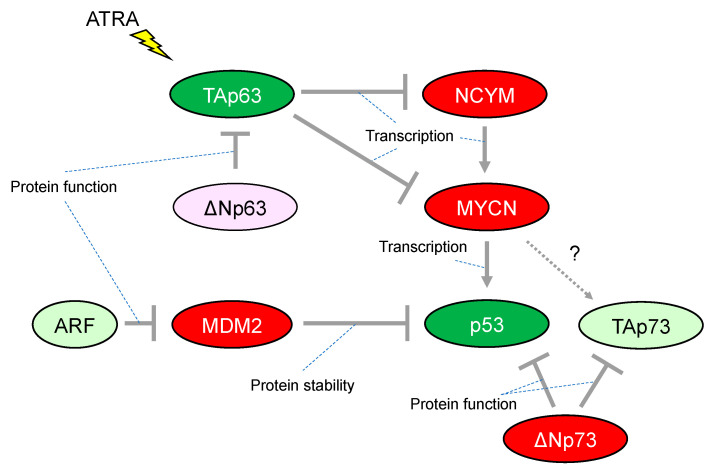
Characteristic alteration of TP53-family-related pathways in neuroblastoma. The red-colored or green-colored molecules indicate oncogenes or tumor-suppressors in neuroblastomas, respectively. The light-pink-colored or light-green-colored molecules could positively or negatively affect neuroblastoma proliferation, respectively. Interaction between TAp73 and MYCN is unknown. ATRA, All-trans-retinoic acid.

**Figure 3 cells-12-00544-f003:**
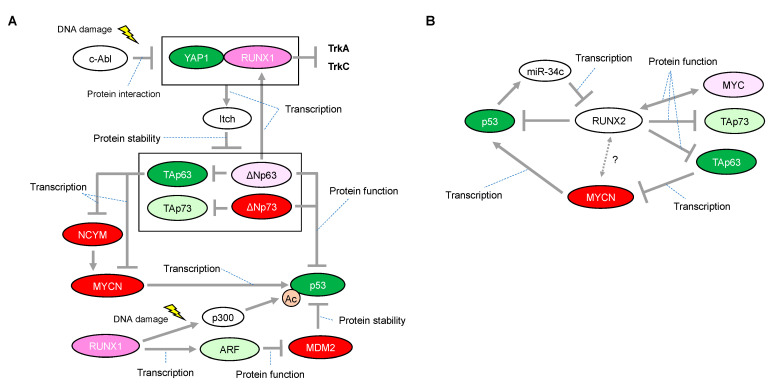
Proposed network of expressional and/or functional regulation between the RUNX1 or RUNX2 and p53 families. (**A**) In response to DNA damage, RUNX1 interacts with Yes-associated protein (Yap1) and collaborates to induce protein degradation of all the p63 and p73 isoforms through the transcriptional activation of Itch, one of the known E3 ligases for p63 and p73. (**B**) Inhibitory roles of RUNX2 on the p53 family. Interaction between RUNX2 and MYCN is unknown. The red-colored or green-colored molecules indicate oncogenes or tumor-suppressors in neuroblastomas, respectively. The pink-colored RUNX1 could positively affect neuroblastoma proliferation The light-pink-colored or light-green-colored molecules could positively or negatively affect neuroblastoma proliferation, respectively, and the non-colored molecules are undetermined.

**Figure 4 cells-12-00544-f004:**
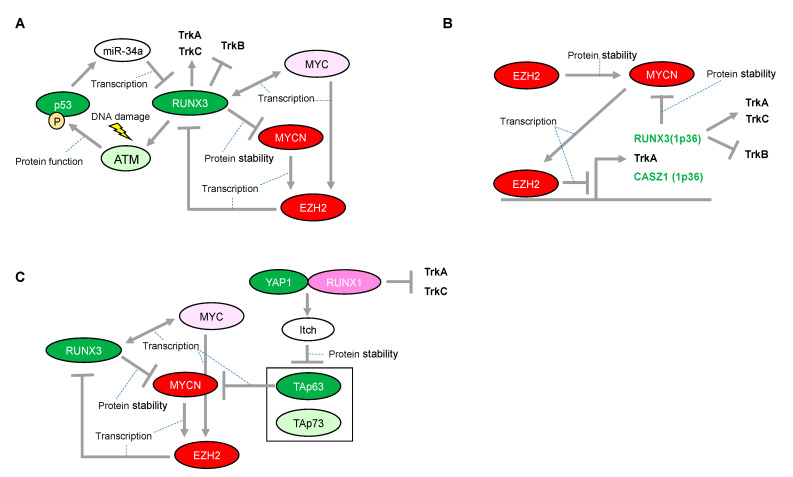
Proposed network between RUNX3 and MYCN in neuroblastomas with presence or absence of functional p53. (**A**) Expressional and/or functional regulation between the RUNX3 and p53 families. (**B**) Enhancer of zeste 2 (EZH2)-mediated epigenetic regulation is associated with neuroblastoma aggressiveness through transcriptional repression of RUNX3. (**C**) RUNX1-dependency in p53 non-functional neuroblastomas. The red-colored or green-colored molecules indicate oncogenes or tumor-suppressors in neuroblastomas, respectively. The pink-colored RUNX1 could positively affect neuroblastoma proliferation. The light-pink-colored or light-green-colored molecules could positively or negatively affect neuroblastoma proliferation, respectively, and the non-colored molecules are undetermined.

**Table 1 cells-12-00544-t001:** Ligands and gene alterations of Trk family in various tumors.

	TrkA	TrkB	TrkC
**Ligand**	NGF	BDNF, NT4/5	NT3
**Mutation**	melanoma, neuroblastoma, acute myeloid leukaemia	colorectal cancer, lung adenocarcinoma, melanoma, large cell neuroendocrine carcinomas (LCNEC)	pancreatic cancer, breast cancer, lung cancer, gastric cancer
**Overexpression**	neuroblastoma (favorable), breast cancer, pheochromocytoma, pancreatic cancer, ovarian cancer, esophageal cancer, bladder cancer	pancreatic adenocarcinoma, prostate cancer, Wilms’ tumor, Hodgkin lymphomas, multiple myeloma, cylindroma, spiradenoma	neuroblastoma (favorable), glioma, medulloblastomas, non-cerebellar primitive neuroectodermal (PNET) tumors, breast cancer, hepatocellular carcinoma, gastric cancer, thyroid cancer, melanoma, leukemia, cylindroma, spiradenoma, adenoid cystic carcinoma
**Gene fusion**	lung adenocarcinoma, intrahepatic cholangiocarcinoma, colorectal cancer, papillary thyroid cancer, spitzoid neoplasm, glioblastoma, sarcoma	astrocytoma, lung adenocarcinoma, head and neck squamous cell carcinoma, brain lower grade glioma	secretory breast carcinoma, mammary analogue secretory carcinoma, papillary thyroid cancer, acute myeloid leukemia, congenital fibrosarcoma, Ph-like acute lymphoblastic leukemia, colon adenocarcinoma, thyroid carcinoma, skin cutaneous melanoma, head and neck squamous cell carcinoma

## Data Availability

Not applicable.

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
