# Peer review of "The RUNX Family Defines Trk Phenotype and Aggressiveness of Human Neuroblastoma through Regulation of p53 and MYCN"

_cells, 2023, doi:10.3390/cells12040544_

Round 1

Reviewer 1 Report

In the article "RUNX family defines Trk phenotype and aggressiveness of human neuroblastoma through regulation of p53 and MYCN" the authors make a complete bibliographic review of the role and relationship between the RUNX family, the TP53 family and the Trk family in neuroblastoma, one of the most frequent childhood tumours. 

The article is well written and the citations used are correct. The content is interesting and may be useful for other researchers in the field. 

However, there are times when it is difficult to follow the text if you are not fully aware of the signalling pathways. Perhaps the authors could make more use of the figures to help you follow along. 

Even so, the article, in my opinion, is ready for publication.

Author Response

Dear Reviewer 1,   We very much appreciate for valuable comments on our revised manuscript from you. As your suggestion, we divided the original Figure 3 into 3 parts based on each RUNX family members for showing the detail interaction as the revised Figure3A, 3B, and Figure4A, and newly added Figure 4C especially for showing speculated a dominant pathway in the p53 non-functional neuroblastomas. We hope that the revisions of our work are satisfactory.   Sincerely yours, Kiyohiro Ando and Akira Nakagawara

Reviewer 2 Report

RUNX family defines Trk phenotype and aggressiveness of human neuroblastoma through regulation of p53 and MYCN 

The abstract is well written

The Runt-related transcription factor (RUNX) family, which has essential roles in neural development and differentiation through regulation of the tropomyosin-related kinase (Trk) family, also has both oncogenic and tumor suppressive roles in various malignant neoplasms through regulation of the p53 family.  Please explain how and by what mechanism the RUNX family regulate the Trk family, please expand on the mechanism and give specific examples. If they appear in the next chapters, then kindly explain this and set the scene for the reader of the next chapters.

Please give more intro to the RUNX family in chapter 1.

The TrK receptors and their ligands have been mentioned and the authors mention that some of the TrK receptors are associated with favourable or unfavourable prognoses, it would be interesting to either expand on this in the text or in a table, so the reader can appreciate that contrasting roles exist amongst this family. This makes sense since you then move on to some examples of Trk translocations and roles in differentiation so such a table will be very helpful.

Figure 1 appears before it has been mentioned in the main text, please move this.

In summary, considering an increase in TrkA and TrkC expression is associated with favorable biological features in neuroblastoma [6,7], RUNX3 may simply be categorized as a functional tumor suppressor in neuroblastoma, because of its potential to positively regulate TrkA and TrkC. In contrast, RUNX1-mediated downregulation of TrkA and TrkC implies that RUNX1 has oncogenic potential in neuroblastomas. Consistently, loss of RUNX3 was associated with an increase in TrkB signaling, which marks an unfavorable prognosis in patients with neuroblastoma [19]. This is an interesting analysis in that depending on the regulating factors these receptors could have contrasting effects in this cancer. Does RUNX1 have any effect on TrkB? Are there any examples which defy this general observation (for example RUNX3 regulation of TrkA and TrkC leading to an unfavourable outcome perhaps in other cancers? 

Consistently, MYCN binds directly to a 145 noncanonical E-Box DNA binding motif located upstream of the transcriptional start site within the p53 promoter and upregulates p53 transcripts in neuroblastoma (Figure 2).  Therefore, MYCN-induced p53 is recognized as a safeguarding mechanism preventing aberrant proliferation in the normal embryonic development of the nervous system [25]. Indeed, this is an example of how oncogenes such as MYCN have a failsafe (built-in negative regulatory mechanism) to reduce their oncogenic spectrum. This also applies to RAS. This is the way the combinatorial effect of MYC and RAS mutually cancel out eachothers dampening mechanisms and increase their combined oncogenic effects.

However, in contrast to p53-deficient mice, those lacking p73 do not show increased susceptibility to spontaneous tumorigenesis, but exhibit neurological and immunological defects. It is interesting to know more about p73 and how much of tumour suppressor it can be when p53 is non-functional. Also, what is the evolutionary relationship between p73 and p53? 

ΔNp73, which is one of the isoforms of p73 lacking the N-terminal transcriptional activated domain, is recognized as an oncogene, conveying a dominant-negative effect on p53 and p73 isoforms with a functional transactivated domain (hereafter, TAp73) (Figure 2). Notably, ΔNp73 may broaden our understanding of the possible link between p53 inactivation and neuroblastoma tumorigenesis. Has anyone looked at the homology between these proteins? If the truncated p73 is a dominant negative for both, then this must mean high homology.

ΔNp73 inhibits neuronal programmed cell death by blocking the pro-apoptotic function of p53. Does this mean that p73 increases apoptosis?

Tap63, therefore, has an interesting role, since after ATRA exposure this protein is produced although it is inhibiting the oncogene, it is also inhibiting the tumour suppressor (since the oncogene was an activator of the tumour suppressor). Could the authors comment on that? In that regard then perhaps ATRA will have conflicting roles as well. ATRA induces differentiation and this can lead also to cell cycle arrest and cell death as by-products of this mechanism (au lieu to differentiation).

Athymic nude mice bearing xenografts formed by subcutaneous injection of neuroblastoma SH-SY5Y cells stably expressing RUNX1 or short hairpin RNAs of RUNX1 displayed inhibited or accelerated tumor development, respectively. These findings are consistent with the reports that RUNX1 overexpression induces cell death, as well as cell cycle arrest or differentiation in neuroblastoma cell lines. Again, this is a spectrum of scenarios that one would expect from differentiation inducers.

In addition, excessive expression of RUNX1 under certain experimental conditions may functionally mimic tumor-suppressive RUNX3 (as described later), depending on the similarity of the consensus sequence for DNA binding, indicating the presence of artificial effects [37].  What do the authors mean by artificial effects?

Therefore, it is plausible that increased expression of RUNX1 enables neuroblastoma cells, which possess functional wild-type p53, to undergo apoptosis in response to a chemotherapeutic agent (Figure 3). However, there is conflicting evidence that, in response to DNA damage, RUNX1 is associated with stabilization of ΔNp73 (as described later), which has the ability to eliminate p53-mediated neuronal apoptosis [30] (Figure 3). Therefore, the dominant function of RUNX1 in neuroblastoma during systematic administration of chemotherapy to patients remains enigmatic. So, the tumour-suppressor role of RUNX1 is context-dependent? Many of chemotherapeutic agents induce DNA damage, so it is strange that DNA damage in the second scenario leads to the opposite effect. Could this be a similar phenomenon to the failsafe effect for an oncogene which negatively regulates its function but now for a tumour suppressor? How were these DNA damage scenarios different?

Regarding the direct activation of p53 by RUNX1 in figure 3 but its indirect activation through suppressor of the inhibitor (ATF blocking MDM2 which blocked p53), while RUNX3 activates ATM is interesting.

In figure 3 you could show the direct activity of MYCN on p53.

In figure 3 please explain the link between the YAP1/RUNX1 complex and p53. The figure shows this complex induces itch, while Itch blocks this box, whereas the authors are stating that itch suppresses deltaNp73 which per se suppresses p53. Please show this with arrows etc. At present, the role of itch in the figure is not that clear. 

The role of RUNX2 is also controversial (based on figure 3), since it is suppressing p53 but also induces MYC, which activates RUNX3/ ATM and p53 in a sequential manner. But also, there may be a link with MYCN which ultimately suppresses the RUNX3/ ATM and p53 axis. So, in fact, this transcription factor cannot be easily assigned a tumour suppressor or oncogene role.

The authors have talked about the role of p53 in the networks of figure 3 but what if the NB cell line is p53 non-functional or MYCN nonamplified? How will this network change? The authors could either explain this in the text or propose a speculative figure.

Figure 3 may get even more complex if Trks get added to it (akin to figure 4).

Thanks

Author Response

Dear Reviewer 2,

We very much appreciate for valuable comments on our revised manuscript from you. For convenience for our responses to the comments, we divided the comments into 11 parts and sincerely responded them one-by-one, and edited the revised manuscript. We hope that the explanations and revisions of our work are satisfactory, and that the manuscript.

Sincerely yours,

Kiyohiro Ando and Akira Nakagawara

Comment 1.

The abstract is well written. The Runt-related transcription factor (RUNX) family, which has essential roles in neural development and differentiation through regulation of the tropomyosin-related kinase (Trk) family, also has both oncogenic and tumor suppressive roles in various malignant neoplasms through regulation of the p53 family.  Please explain how and by what mechanism the RUNX family regulate the Trk family, please expand on the mechanism and give specific examples. If they appear in the next chapters, then kindly explain this and set the scene for the reader of the next chapters. Please give more intro to the RUNX family in chapter 1.

Response 1.

We quite agree with your suggestion that we should introduce an association RUNX family and Trk family in introduction section for setting the scene for the readers of the following chapter. We added explanation in the introduction section as follows:

(Page 2, line 46) Knockout mice studies have indicated that Runx1 down-regulates the number of both TrkA- and TrkC- positive neurons. In contrast, Runx3 up-regulates the number of both TrkA- and TrkC-positive neurons through repression of TrkB signaling (reviewed in [6] ).

Comment 2.

The TrK receptors and their ligands have been mentioned and the authors mention that some of the TrK receptors are associated with favourable or unfavourable prognoses, it would be interesting to either expand on this in the text or in a table, so the reader can appreciate that contrasting roles exist amongst this family. This makes sense since you then move on to some examples of Trk translocations and roles in differentiation so such a table will be very helpful.

Figure 1 appears before it has been mentioned in the main text, please move this.

Response 2.

We understand your suggestion that role of Trk family overexpression and mutations in promoting tumorigenesis and cancer development has been reported in multiple tumors, therefore we should summarize it as a table. We removed Figure 1 appropriately, then newly added Table 1, and modified the revised text with an additional reference as follows:

(Page 2, line 85) Somatic mutations of the Trk family have been identified in various tumor types. However, the role of each Trk overexpression and mutations in promoting tumorigenesis and cancer progression has been reported as variable with either accelerating or inhib-iting effects. Most notably, there is increasing evidence that oncogenic translocation of Trk family members is associated with morbidity and mortality, as well as clinical efficacy of targeted therapeutics (e.g., NTRK inhibitors) in multiple tumors from both adult and pediatric patients (Table 1) [23-25].

(An additional reference)

  1. Jin, W. Roles of TrkC Signaling in the Regulation of Tumorigenicity and Metastasis of Cancer. Cancers 2020, 12, doi:10.3390/cancers12010147.

Comment 3.

In summary, considering an increase in TrkA and TrkC expression is associated with favorable biological features in neuroblastoma [6,7], RUNX3 may simply be categorized as a functional tumor suppressor in neuroblastoma, because of its potential to positively regulate TrkA and TrkC. In contrast, RUNX1-mediated downregulation of TrkA and TrkC implies that RUNX1 has oncogenic potential in neuroblastomas. Consistently, loss of RUNX3 was associated with an increase in TrkB signaling, which marks an unfavorable prognosis in patients with neuroblastoma [19]. This is an interesting analysis in that depending on the regulating factors these receptors could have contrasting effects in this cancer. Does RUNX1 have any effect on TrkB? Are there any examples which defy this general observation (for example RUNX3 regulation of TrkA and TrkC leading to an unfavourable outcome perhaps in other cancers?

Response 3.

We understand your concern whether TrkB expression could be regulated by Runx family. In this regard, Kramer and colleagues reported that Runx1 did not induce expression of TrkB. It has been suggested that TrkB expression in prospective proprioceptors might be counteracted by Runx3 expression. We added the reference in the revised text as follows:

(Page 4, line 129) Subsequent studies demonstrated that RUNX3 determines TrkC-positive neuron iden-tities in the presence of NT3 (the ligand for TrkC) through the transcriptional repression of TrkB, in order to segregate from TrkBTrkC double positive neurons [30,31].

(An additional reference)

  1. Kramer, I.; Sigrist, M.; de Nooij, J.C.; Taniuchi, I.; Jessell, T.M.; Arber, S. A role for Runx transcription factor signaling in dorsal root ganglion sensory neuron diversification. Neuron 2006, 49, 379-393, doi:10.1016/j.neuron.2006.01.008.

In relation to your Comment 2, we also agree your suggestion and should describe another aspect of TrkA and TrkC function in various cancers that you pointed out. An autocrine loop involving TrkA and NGF is associated with protumorigenic activity in both breast and prostate carcinomas (Dolle et al. and Donato et al, respectively). Similar to the TrkA, a subset of stage IV neuroblastomas exhibits increased co-expression of NT-3 and TrkC, and high-grade gliomas showed a more positive immunoreactivity than low-grade gliomas in NT-3 and TrkC expression (Bouzas et al. and Calatozzolo et al, respectively). We described these in the revised text with additional references.

(Page 2, line 80) In contrast, an autocrine loop of the NGF/TrkA is associated with protumorigenic activity in both breast and prostate carcinomas [18,19]. Similarly, increased co-expression of NT-3 and TrkC are observed in a subset of stage IV neuroblastomas and high-grade gliomas [20,21]. These findings are possibly linked with the idea that TrkA and TrkC behave as dependence receptors, which trigger apoptosis when unbound by their respective ligands [22].  

(Additional references)

  1. Di Donato, M.; Cernera, G.; Migliaccio, A.; Castoria, G. Nerve Growth Factor Induces Proliferation and Aggressiveness In Prostate Cancer Cells. Cancers 2019, 11, doi:10.3390/cancers11060784.
  2. Dolle, L.; Adriaenssens, E.; El Yazidi-Belkoura, I.; Le Bourhis, X.; Nurcombe, V.; Hondermarck, H. Nerve growth factor receptors and signaling in breast cancer. Curr Cancer Drug Targets 2004, 4, 463-470, doi:10.2174/1568009043332853.
  3. Bouzas-Rodriguez, J.; Cabrera, J.R.; Delloye-Bourgeois, C.; Ichim, G.; Delcros, J.G.; Raquin, M.A.; Rousseau, R.; Combaret, V.; Benard, J.; Tauszig-Delamasure, S.; et al. Neurotrophin-3 production promotes human neuroblastoma cell survival by inhibiting TrkC-induced apoptosis. J Clin Invest 2010, 120, 850-858, doi:10.1172/JCI41013.
  4. Calatozzolo, C.; Salmaggi, A.; Pollo, B.; Sciacca, F.L.; Lorenzetti, M.; Franzini, A.; Boiardi, A.; Broggi, G.; Marras, C. Expression of cannabinoid receptors and neurotrophins in human gliomas. Neurol Sci 2007, 28, 304-310, doi:10.1007/s10072-007-0843-8.
  5. Nikoletopoulou, V.; Lickert, H.; Frade, J.M.; Rencurel, C.; Giallonardo, P.; Zhang, L.; Bibel, M.; Barde, Y.A. Neurotrophin receptors TrkA and TrkC cause neuronal death whereas TrkB does not. Nature 2010, 467, 59-63, doi:10.1038/nature09336.

Comment 4.

Consistently, MYCN binds directly to a noncanonical E-Box DNA binding motif located upstream of the transcriptional start site within the p53 promoter and upregulates p53 transcripts in neuroblastoma (Figure 2).  Therefore, MYCN-induced p53 is recognized as a safeguarding mechanism preventing aberrant proliferation in the normal embryonic development of the nervous system [25]. Indeed, this is an example of how oncogenes such as MYCN have a failsafe (built-in negative regulatory mechanism) to reduce their oncogenic spectrum. This also applies to RAS. This is the way the combinatorial effect of MYC and RAS mutually cancel out eachothers dampening mechanisms and increase their combined oncogenic effects.

Response 4.

We quite agree with your comment, thus we added the explanation in the revised text with additional reference as follows:

(Page 5, line 181) It is quite similar to the observation that reintroduction of oncogenic RAS in primary human or rodent cells results in cellular senescence through accumulation of p53. [38] In addition, the transformation from rodent fibroblasts does not occur with transfection of Ras oncogenes alone, unless a cooperating oncogene such as Myc, another member of the MYC family (of MYCN) [39].

(Additional references)

  1. Serrano, M.; Lin, A.W.; McCurrach, M.E.; Beach, D.; Lowe, S.W. Oncogenic ras provokes premature cell senescence associated with accumulation of p53 and p16INK4a. Cell 1997, 88, 593-602, doi:10.1016/s0092-8674(00)81902-9.
  2. Land, H.; Parada, L.F.; Weinberg, R.A. Tumorigenic conversion of primary embryo fibroblasts requires at least two cooperating oncogenes. Nature 1983, 304, 596-602, doi:10.1038/304596a0.

Comment 5.

However, in contrast to p53-deficient mice, those lacking p73 do not show increased susceptibility to spontaneous tumorigenesis, but exhibit neurological and immunological defects. It is interesting to know more about p73 and how much of tumour suppressor it can be when p53 is non-functional. Also, what is the evolutionary relationship between p73 and p53?

ΔNp73, which is one of the isoforms of p73 lacking the N-terminal transcriptional activated domain, is recognized as an oncogene, conveying a dominant-negative effect on p53 and p73 isoforms with a functional transactivated domain (hereafter, TAp73) (Figure 2). Notably, ΔNp73 may broaden our understanding of the possible link between p53 inactivation and neuroblastoma tumorigenesis. Has anyone looked at the homology between these proteins? If the truncated p73 is a dominant negative for both, then this must mean high homology.

ΔNp73 inhibits neuronal programmed cell death by blocking the pro-apoptotic function of p53. Does this mean that p73 increases apoptosis?

Response 5.

We apologize that explanation for the overviews of p73 and p63 are insufficient in the text. Based on your concerns, we revised the text including in the introduction section with an additional reference as follows:

(Page 2, line 49) The p53 family members, p63 and p73, possess three major functional domains composed by the NH2-terminal transactivation domain, the central core sequence-specific DNA-binding domain, and the COOH-terminal oligomerization domain. A study from mammalian oocytes suggests that p63 is the most ancient form, then evolve to p73, which is structurally more similar to p63 than p53. Later in evolution, p53 appeared in somatic cells as a tumor suppressor [7]. There are two major NH2-terminally distinct isoforms as transcriptionally active (TA) and transcriptionally inactive (∆N) forms. Of note, the DNA-binding domain is highly conserved across the family. Therefore, p63 and p73 share the majority of the p53-responsive promoters, and have an ability to transactivate a series of p53-target genes implicated in cell cycle arrest and apoptosis (reviewed in [8]). All of RUNX family member interact with p53 and can influence tumor-suppressive ability [9], whereas RUNX1 contributes to induce protein degradation of p63 and p73 [10,11].

(Additional references)

  1. Dotsch, V.; Bernassola, F.; Coutandin, D.; Candi, E.; Melino, G. p63 and p73, the ancestors of p53. Cold Spring Harb Perspect Biol 2010, 2, a004887, doi:10.1101/cshperspect.a004887.
  2. Ozaki, T.; Nakagawara, A. p73, a sophisticated p53 family member in the cancer world. Cancer science 2005, 96, 729-737, doi:10.1111/j.1349-7006.2005.00116.x.
  3. Ozaki, T.; Nakagawara, A.; Nagase, H. RUNX Family Participates in the Regulation of p53-Dependent DNA Damage Response. Int J Genomics 2013, 2013, 271347, doi:10.1155/2013/271347.

(Page 5, line 205) Activation of p73 isoforms with a functional transactivated domain (hereafter, TAp73) in human cancer in response to chemotherapeutic regents induce apoptosis even in the cells lacking p53. A p53 mutant protein can interact with TAp73, resulting in an impaired ability for apoptotic induction [43].

(An additional reference)

  1. Irwin, M.S.; Kondo, K.; Marin, M.C.; Cheng, L.S.; Hahn, W.C.; Kaelin, W.G., Jr. Chemosensitivity linked to p73 function. Cancer cell 2003, 3, 403-410, doi:10.1016/s1535-6108(03)00078-3.

Comment 6.

Tap63, therefore, has an interesting role, since after ATRA exposure this protein is produced although it is inhibiting the oncogene, it is also inhibiting the tumour suppressor (since the oncogene was an activator of the tumour suppressor). Could the authors comment on that? In that regard then perhaps ATRA will have conflicting roles as well. ATRA induces differentiation and this can lead also to cell cycle arrest and cell death as by-products of this mechanism (au lieu to differentiation).

Response 6.

We quite agree with your comment, we should mention a differential role of TAp63 in response to ATRA treatment as compared to p53. In our knowledge, ATRA treatment induce TAp63 expression leading repression of MYCN expression. Subsequently, the down-regulation of MYCN results in inhibiting expression of its transcriptional targets including tumor suppressor p53. Therefore, differential roles on regulating MYCN expression by p53 family may be causally related to the conflicting cellular responses to ATRA treatment in neuroblastomas. We added the explanation in the revised text as follows:

(Page 6, line 234) Therefore, it has been proposed that ATRA treatment induce TAp63 expression leading transcriptional repression of MYCN expression. Subsequently, the down-regulation of MYCN results in inhibiting expression of its transcriptional targets including tumor suppressor p53, indicating that differential functions among p53 family may be causally related to the conflicting cellular responses to ATRA treatment such as promoting neuronal differentiation and/or apoptosis in neuroblastomas.

Comment 7.

Athymic nude mice bearing xenografts formed by subcutaneous injection of neuroblastoma SH-SY5Y cells stably expressing RUNX1 or short hairpin RNAs of RUNX1 displayed inhibited or accelerated tumor development, respectively. These findings are consistent with the reports that RUNX1 overexpression induces cell death, as well as cell cycle arrest or differentiation in neuroblastoma cell lines. Again, this is a spectrum of scenarios that one would expect from differentiation inducers.

In addition, excessive expression of RUNX1 under certain experimental conditions may functionally mimic tumor-suppressive RUNX3 (as described later), depending on the similarity of the consensus sequence for DNA binding, indicating the presence of artificial effects [37].  What do the authors mean by artificial effects?

Therefore, it is plausible that increased expression of RUNX1 enables neuroblastoma cells, which possess functional wild-type p53, to undergo apoptosis in response to a chemotherapeutic agent (Figure 3). However, there is conflicting evidence that, in response to DNA damage, RUNX1 is associated with stabilization of ΔNp73 (as described later), which has the ability to eliminate p53-mediated neuronal apoptosis [30] (Figure 3). Therefore, the dominant function of RUNX1 in neuroblastoma during systematic administration of chemotherapy to patients remains enigmatic. So, the tumour-suppressor role of RUNX1 is context-dependent? Many of chemotherapeutic agents induce DNA damage, so it is strange that DNA damage in the second scenario leads to the opposite effect. Could this be a similar phenomenon to the failsafe effect for an oncogene which negatively regulates its function but now for a tumour suppressor? How were these DNA damage scenarios different?

Response 7.

We quite agree with your comments that the RUNX1-mediated differentiation as well as inducing cell cycle arrest and apoptosis are recognized as a spectrum of scenarios. Therefore, we revised the text to “suggesting the presence of functional redundancy” from “indicating the presence of artificial effects” (page 7 at line 277). In line with this spectrum scenarios, it is reasonable that the chemotherapeutic agents-induced p53 activation is accomplished, at least in part, through the cooperation with RUNX1, while RUNX1-mediated stabilization of ΔNp73 is immediately initiated as a fail-safe mechanism to prevent detrimental effects caused by the excessive p53 activation. Therefore, as you mentioned, we revised the text to describe that the stabilization of ΔNp73 is one of the nature of fail-safe mechanism in cells as follows: 

(Page 8, line 312) Taken together with these findings, it is reasonable that the chemotherapeutic agents-induced p53 activation is accomplished, at least in part, through the cooperation with RUNX1, while RUNX1-mediated stabilization of ΔNp73 is immediately initiated as a fail-safe mechanism to prevent detrimental effects caused by the excessive p53 activation.

Comment 8.

Regarding the direct activation of p53 by RUNX1 in figure 3 but its indirect activation through suppressor of the inhibitor (ARF blocking MDM2 which blocked p53), while RUNX3 activates ATM is interesting.

In figure 3 you could show the direct activity of MYCN on p53.

In figure 3 please explain the link between the YAP1/RUNX1 complex and p53. The figure shows this complex induces itch, while Itch blocks this box, whereas the authors are stating that itch suppresses deltaNp73 which per se suppresses p53. Please show this with arrows etc. At present, the role of itch in the figure is not that clear.

Response 8.

As you mentioned, it is interesting that RUNX1 contributes not only to enhance activity of p53 through ARF and ATM, but also to repress p53 function through ΔNp73/ΔNp63.

We agree that we should indicate the direct activity of MYCN on p53 in Figure 3. However, due to its complexity of representation all together, we divided the original Figure 3 into 3 parts based on each RUNX family members showing as the revised Figure3A, 3B, and Figure4A. We also added the explanation that the link of YAP1/RUNX, and the roles of itch for the p73 and p63 protein degradation in a legend of the revised Figure 3 as follows:

(Revised the legend of Figure 3) A, In response to DNA damage, RUNX1 interacts with Yes-associated protein (Yap1) and collaborates to induce protein degradation of all the p63 and p73 isoforms through the transcriptional activation of Itch, one of the known E3 ligases for p63 and p73.

Comment 9.

The role of RUNX2 is also controversial (based on figure 3), since it is suppressing p53 but also induces MYC, which activates RUNX3/ ATM and p53 in a sequential manner. But also, there may be a link with MYCN which ultimately suppresses the RUNX3/ ATM and p53 axis. So, in fact, this transcription factor cannot be easily assigned a tumour suppressor or oncogene role.

Response 9.

We understand your concern that Runx2 neutralizes the pro-apoptotic effects of the Myc overexpression possibly through repression of p53 function. Whereas MYC can induce RUNX3 transcripts to counteract the RUNX2-mediated inhibition of p53 function, it is plausible that there may exist a mechanism to inhibit the RUNX3/p53 axis. As you mentioned, one of the possible mechanisms is that MYCN can repress RUNX3 expression through directly inducing EZH2 expression, and repress RUNX3 expression. Interestingly, it has been reported that not only MYCN but also MYC possibly has a ability to induce EZH2 expression, therefore we indicated MYC/EZH axis in the revised Figure 4A and added explanation in the revised text as follows:

(Page 12, line 520) Since both MYCN and MYC directly induces EZH2 expression through binding at its promoter region [96,97], EZH2-mediated inactivation of the tumor suppressors in 1p36 may conceivably be a major route toward development of unfavorable neuroblastomas with MYCN amplification (Figure 4B).

(An additional reference)

  1. Koh, C.M.; Iwata, T.; Zheng, Q.; Bethel, C.; Yegnasubramanian, S.; De Marzo, A.M. Myc enforces overexpression of EZH2 in early prostatic neoplasia via transcriptional and post-transcriptional mechanisms. Oncotarget 2011, 2, 669-683, doi:10.18632/oncotarget.327.

Comment 10.

The authors have talked about the role of p53 in the networks of figure 3 but what if the NB cell line is p53 non-functional or MYCN nonamplified? How will this network change? The authors could either explain this in the text or propose a speculative figure.

Response 10.

We quite agree with your comments that we should propose the models for dominant pathways in p53 non-functional or MYCN nonamplified neuroblastomas for suggesting a future target therapy. In this regard, we added a speculative figure for the dominant pathways in p53 non-functional neuroblastomas as Figure 4C, and summarized prospective pathways including MYCN non-amplified neuroblastomas in the revised discussion section as follows:

(Page 13, line 581) Similarly, in the MYCN-nonamplified neuroblastomas, targeting the RUNX1-mediated p53 activation may be effective for abolishing their fail-safe mechanism, because of presence of other oncogenes. Conversely, the p53 non-functional neuroblastomas may rely on TAp63 and TAp73 for their survival suggesting that the reactivation of RUNX1 is possibly to be a therapeutic option (Figure 4C). Further investigation of targeting RUNX1 may be advantageous as a potential therapeutic strategy for neuroblastoma.

Comment 11.

Figure 3 may get even more complex if Trks get added to it (akin to figure 4).

Thanks

Response 11.

We agree with your comment for better understanding the overview of the pathway. We added the Trk family in the revised Figure 3 and 4.

Round 2

Reviewer 2 Report

My comments have been addressed.